# Identification of a Set of Variables for the Classification of Páramo Soils Using a Nonparametric Model, Remote Sensing, and Organic Carbon

Yadira Pazmiño [1,*], José Juan de Felipe [1], Marc Vallbé [1], Franklin Cargua [2] and Luis Quevedo [3]

1 Department of Mining, Industrial and ICT Engineering, Manresa School of Engineering, Universitat Politècnica de Catalunya, 08242 Manresa, Spain; jose.juan.de.felipe@upc.edu (J.J.d.F.); marc.vallbe@upc.edu (M.V.)
2 Research and Development Group for the Environment and Climate Change, Escuela Superior Politécnica de Chimborazo, Riobamba 060150, Ecuador; franklin.carguac@espoch.edu.ec
3 Tourism Department, Universidad Nacional de Chimborazo UNACH, Riobamba 060150, Ecuador; luis.quevedo@unach.edu.ec
* Correspondence: yadira.pazmino@estudiant.upc.edu

**Abstract:** Páramo ecosystems harbor important biodiversity and provide essential environmental services such as water regulation and carbon sequestration. Unfortunately, the scarcity of information on their land uses makes it difficult to generate sustainable strategies for their conservation. The purpose of this study is to develop a methodology to easily monitor and document the conservation status, degradation rates, and land use changes in the páramo. We analyzed the performance of two nonparametric models (the CART decision tree, CDT, and multivariate adaptive regression curves, MARS) in the páramos of the Chambo sub-basin (Ecuador). We used three types of attributes: digital elevation model (DEM), land use cover (Sentinel 2), and organic carbon content (Global Soil Organic Carbon Map data, GSOC) and a categorical variable, land use. We obtained a set of selected variables which perform well with both models, and which let us monitor the land uses of the páramos. Comparing our results with the last report of the Ecuadorian Ministry of Environment (2012), we found that 9% of the páramo has been lost in the last 8 years.

**Keywords:** páramo; sustainability; land use; predictive nonparametric models; natural conservation; degradation of natural resources; remote sensing

## 1. Introduction

The Andean moorlands (known as páramos) extend along Costa Rica, Colombia, Ecuador, Venezuela, and Perú at heights between 3500 and 5000 m.a.s.l. Most of them are of volcanic origin and have quite complex geology and topography [1]. They provide several ecosystem services such as water supply and regulation, biodiversity conservation, and carbon storage [2]. In fact, Andean moorlands receive between 600 and 1000 mm of precipitation per year, which represents approximately 2/3 of the annual precipitation in the Andes, making them the main providers of water in this region [3]. The upper layers of páramo ecosystems can retain up to 183 tons of carbon/ha [4], which is significant because soil organic carbon (SOC) is considered the largest terrestrial, non-sedimentary carbon reserve [5]. Páramo soils are characterized by being dark and humic with an open and porous structure and present a slow process of organic matter degradation due to the low levels of temperature and atmospheric pressure [6].

Ecuadorian páramos run from 0°49′ N to 4°52′ S and cover approximately 833,834 ha, representing approximately 5% of the country's total area. They are home to a large collection of Neotropical–Alpine ecosystems, containing approximately 628 endemic plant species, which is equivalent to 15% of the endemic flora and 4% of the total flora of the entire country. Forty-eight percent of this flora is located inside protected areas and 75% of

its endemic species are threatened [7]. However, anthropogenic activities such as urban settlements, agriculture, and livestock have intensified throughout history, constantly degrading páramo ecosystems [8,9], resulting in a decrease in carbon dioxide sinks [10,11] and in significant alterations in water regulation, erosion, and drought [12].

In this context, a sustainable management of the páramo becomes essential to balance economic growth, population growth, and environmental protection on one hand [13,14], and those who benefit directly or indirectly from its ecosystem services on the other hand. This challenge requires the implementation of adequate policies that consider the changes in land use, the growing population, the growing demand for agricultural products, the adverse effects on the climate, and the functions of the páramo ecosystem [15].

Unfortunately, the current situation is difficult to assess because there are no thorough databases, especially on the state of the resource itself. Environmental data are hard to obtain due to climatic conditions and difficulty of access. While performing rapid, systematic, economic, and efficient monitoring in hard-to-reach places is crucial for determining sustainability strategies and policies for natural areas, limited access to the páramos makes the sustainable management of this resource particularly challenging [16]. Several studies recommend the use of remote sensing data to generate information in inaccessible areas [17].

Statistical methods for the management of environmental data have two main advantages: the first one is the simplicity of handling and generating large amounts of datasets and the second one is the low cost involved since there are free software tools available for its use. These statistical methods provide excellent results in the production of information from areas with little data available [18,19]. Hence, the combination of geographical information systems (GIS), remote sensing, and statistical models is becoming increasingly popular due to its convenience, low cost, and time-saving features compared to field monitoring [20].

The evaluation of the different land uses through satellite imaging has been widely performed by many authors [21] using different nonparametric statistical techniques, which allow the creation of predictive algorithms from basic remote sensing data [22–24]. Thus, neural networks have been used to determine land use through images from different satellites [25,26] and other authors have used different machine learning algorithms to interpret satellite data [27]. In addition, some studies have evaluated the different land uses by analyzing the signal characteristics from satellite images [28], while other works have created predictive algorithms from nonparametric statistical tools, complementing satellite data with data from other sources. These nonparametric statistical methods use various types of attributes (e.g., topographic and remote data, lithology, and soil properties) [29], which allow to determine the status of categorical variables related to the conservation of natural resources, such as landslides, mineralogy, or the SOC concentration [30,31].

Nonparametric probabilistic (e.g., CDT) and automatic learning methods have been widely exploited for vegetation mapping [32] and ecological modeling [33,34], while the MARS method has been used in studies related to hydrology [35] and geomorphology, and also supporting the creation of algorithms in remote sensing [36].

The objective of this study is to evaluate the performance of a supervised learning algorithm that will allow to know the target variable by learning decision rules used from the characteristics of the response variable to monitor and know the different land uses of the páramo using three types of attributes (topographic data, Sentinel 2 satellite remote data, and SOC data) and a categorical variable (land use coverage). Then, the information generated is used to assess the performance of two nonparametric statistical models: the CART decision tree (CDT) and multivariate adaptive regression splines (MARS). These two statistical models were analyzed to obtain a set of variables that allow for a reliable land use classification.

The information that can be obtained through the proposed methodology contributes to the generation of databases in a fast, economical, and efficient way that monitors the status of the Andean moorlands. These databases can be used as technical inputs for

changes in páramo land use. Together with other social and economic inputs, they become a great tool for conservation strategies and policies that are more in line with the reality of the páramo ecosystem.

## 2. Materials and Methods

### 2.1. Study Area

This study is set on the Andean páramos located in the Chambo sub-basin, which has a population of 414,495 inhabitants and an annual population growth rate of 1.42%. It is located at 78°39′ W and 1°39′ S at an altitude of 3500 m.a.s.l. and bounded by the Chimborazo Faunal Production Reserve and Sangay National Park (Figure 1). It has an area of 3580 km$^2$, of which 42.1% is páramo, and covers 51% of the total area of the province. The main land uses are focused on agricultural and livestock activities [37]. Temperatures in the area can vary between 2 °C and 20 °C according to the National Institute of Meteorology and Hydrology of Ecuador.

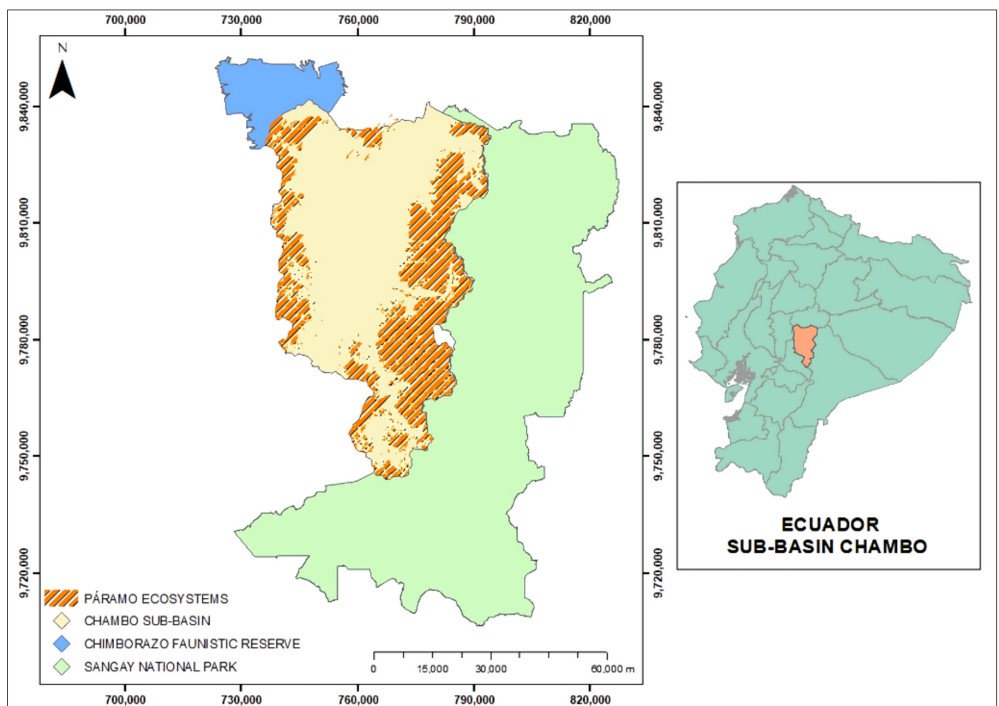

**Figure 1.** Location of the study area.

### 2.2. Work Flow

The study was carried out through the following stages: image processing, segmentation, variable selection, value extraction, evaluation of the supervised learning models CDT and MARS, and validation of the studied model (Figure 2).

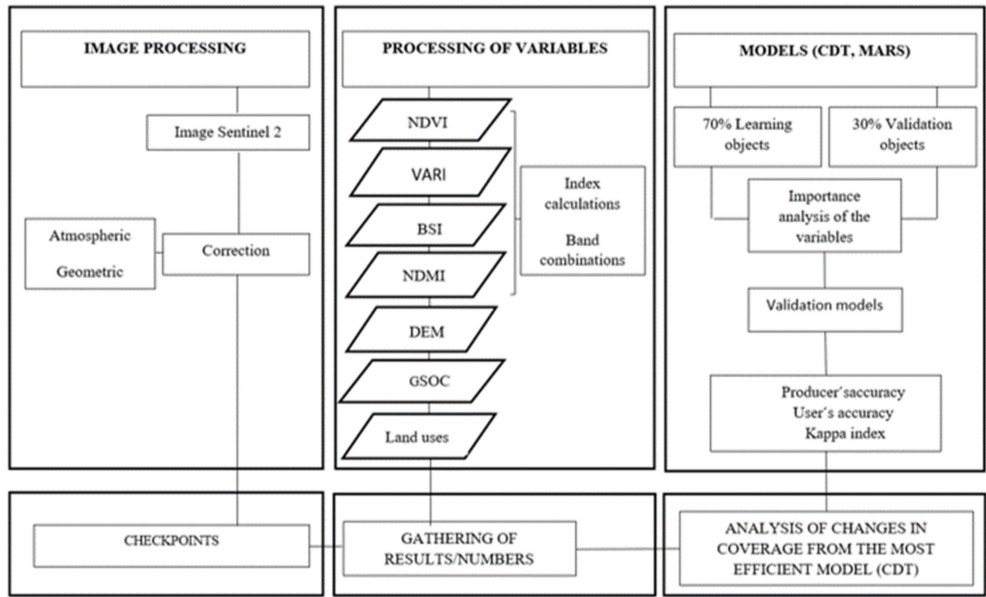

**Figure 2.** Flow chart: main stages of the study.

*2.3. Satellite Images*

Sentinel 2 images were downloaded from the European Space Agency (ESA) through the official Copernicus Open Access Center. Sentinel's Scientific Data Hub user interface was used to set the cloudiness below 30% with the purpose of making the visible field as wide as possible. Twenty satellite images were used for the study area, creating a mosaic (60 km × 60 km). The images corresponded to the years 2019 and 2020.

*2.4. Image Processing*

Atmospheric correction was carried out with the free application Sen2Cor v2.8 of the ESA, through which the bands of the original images were converted from level 1C aerosol optical thickness (AOT) to level 2A bottom of atmosphere (BOA), also known as surface reflectance. The AOT was treated by eliminating the contained water vapor and the images were corrected according to the bands of the lower part of the atmosphere, for which the bands were 12 (SWIR), 4 (red), and 2 (blue) [38].

The geometric correction of the images was carried out by comparing the vector layers of the topographic maps with base maps of rivers and roads corresponding to the Ecuadorian Military Geographic Institute by means of QGIS software [39].

*2.5. Checkpoints*

Through random sampling using QGIS software [40], one point per 100 ha of land was extracted from the official Map of Ecosystems of Continental Ecuador (2012). This resulted in a total of 3580 random control points that we considered adequate for statistical purposes. An additional 20% of points were entered for which there are official data from the Ministry of Agriculture and Livestock. As a last step, 381 points were eliminated because they were located in areas where the Sentinel 2 images had pixels with noise due to atmospheric effects or did not coincide with the coverages observed in the image. After this selection, we were left with a total of 3915 control points that we used in the study, of which a proportional 70% of the sample was verified in situ.

At the end, the points were counted for each coverage studied and its distribution was as follows: 700 points for crops (C), 1290 for pastures (Gr), 150 for forest plantations (PF), 1600 for páramo (Pr), and 175 for areas without vegetation (bare soil, S). These categories correspond to the main land uses according to the report "Contribution to the integrated management of water resources planning" [41].

### 2.6. Variables

We experimented with thirteen variables that characterize the area: the digital elevation model (DEM), taken from the databases of the Ecuadorian Military Geographic Institute [39]; the GSOC, which was extracted from the Global Soil Organic Carbon Map data [42]; and ten spectral indices (Table 1), which were calculated though an algebra map expression using the Python syntax by QGIS 3.4 (2018) based on the combination of corresponding bands according to the methodology by the PyQGIS Programmer's Guide [40]. The categorical variable was the land uses of the páramo ecosystem. The types categorized included C, Gr, PF, Pr, and S. Accuracy was assessed with the statistical measures described in Table 2.

Categorical and topographic variables were selected from the studies that showed the advance of the agricultural frontier as one of the main causes for páramo loss [43,44]. The GSOC and spectral indices were included since they are directly related to the different soil surfaces in a particular way for each of the covers studied; thus, they contribute significantly to the model fitting.

**Table 1.** Spectral indices analyzed from the Sentinel 2 bands.

| Index | Formula | Characteristics |
|---|---|---|
| **NDVI**: Normalized difference vegetation index | $\text{NDVI} = \frac{(\text{NIR}-\text{RED})}{(\text{NIR}+\text{RED})}$ | Minimizes topographic effects and produces a linear measurement scale. Negative values represent areas without vegetation. The higher the index is, the higher the chlorophyll index is [45]. |
| **SAVI**: Soil-adjusted vegetation index | $\text{SAVI} = \frac{(\text{NIR}-\text{RED})}{(\text{NIR}+0.428)\times(1.428)}$ | Minimizes the effect of the soil in areas with low vegetation density [46]. |
| **VARI**: Visible atmospherically resistant index | $\text{VARI} = \frac{(\text{GREEN}-\text{RED})}{(\text{GREEN}+\text{RED}-\text{BLUE})}$ | It highlights vegetation in the visible part of the spectrum, while mitigating differences in lighting and atmospheric effects [47]. |
| **EVI**: Improved vegetation index | $\text{EVI} = 2.5 \times \frac{(\text{NIR}-\text{RED})}{((\text{NIR}+6\times\text{RED}-7.5\times\text{BLUE})+1)}$ | It corrects some atmospheric conditions, e.g., the background noise of the canopy, and it is more sensitive in areas with dense vegetation [48]. |
| **BSI**: Bare soil index | $\text{BSI} = \frac{[(\text{SWIR}+\text{RED})-(\text{NIR}+\text{BLUE})]}{[(\text{SWIR}+\text{RED})+(\text{NIR}+\text{BLUE})]}$ | The difference in the number of areas of bare soil, land, and vegetation [49]. |
| **NGRDI**: Normalized red green difference index | $\text{NGRDI} = \frac{(\text{GREEN}-\text{RED})}{(\text{GREEN}+\text{RED})}$ | Reflectance of the green and red area of the electromagnetic spectrum, which come from a true color image [49]. |
| **ARVI**: Atmospheric resistant vegetation index | $\text{ARVI} = \frac{(\text{NIR}-(2\times\text{Red})+\text{Blue})}{(\text{NIR}+(2\times\text{Red})+\text{Blue})}$ | Recommended for areas with a high concentration of some type of aerosol, mist, smoke, or other type of particles suspended in the air [50]. |

**Table 1.** *Cont.*

| Index | Formula | Characteristics |
|---|---|---|
| **GCI**: Green coverage index | $GCI = \frac{B9}{GREEN} - 1$ | It can specify the health status of the vegetation or warn of the start of temporary seasons [47]. |
| **GNDVI**: Green normalized difference vegetation index | $GNDVI = \frac{(NIR - GREEN)}{(GREEN + NIR)}$ | It is a measure of the "greenness" of the plant or photosynthetic activity. This index is mainly used in the intermediate and final stages of the crop cycle [45]. |
| **NDMI**: Normalized difference moisture index | $NDMI = \frac{(NIR - SWRI)}{(NIR + SWRI)}$ | It describes the level of water stress of the vegetation and between the difference and the sum of the radiation refracted in the near-infrared and SWIR [51]. |

NIR: Near-infrared; RED: band 4; GREEN: band 3; BLUE: band 2; SWRI: band 11; WATER VAPOUR: band 9.

**Table 2.** Statistical values used to evaluate the performance of the predictive models analyzed and the precision of the land-use map.

| Measurement | Formula-Defines Each Parameter in the Description | Description |
|---|---|---|
| Producer's accuracy (PA) | $PA = \frac{D_{ij}}{R_J}$ | Producer's accuracy is a reference-based accuracy that is computed by reviewing the predictions produced by a class and by establishing the percentage of correct predictions [52]. |
| User's accuracy (UA) | $UA = \frac{D_{ij}}{C_i}$ | User's accuracy is a map-based accuracy that is computed by reviewing the reference data for a class and establishing the percentage of correct predictions for these samples [53]. |
| Overall accuracy (OA) | $UA = \frac{\sum D_{ij}}{N}$ | Indicates the proportion of all reference pixels that are correctly classified [53]. |
| Kappa index | $K = \frac{Pr_{(a)} - Pr_{(e)}}{1 - Pr_{(e)}}$ | Concordance between the observed values of the image and the values estimated by the classifier [13]. |
| Indicators of change Gain Losses Net change | Gain $(G_{ij}) = P_{+j} - P_{jj}$ <br> Losses $(L_{ij}) = P_{j+} - P_{jj}$ <br> Net Change $(D_j) = l\, L_{ij} - G_{ij}\, l$ <br> Total change $(D_{TJ}) = G_{ij} + L_{ij}$ <br> Exchange $(S_j) = 2 \times MIN\,(P_{j+} - P_{jj}, P_{+j} - P_{jj})$ | They make it possible to determine for each category gains, losses, net change, and exchanges experienced between two points in time [54]. |
| Systematic transitions in terms of gain and loss | $G_{ij}\, \dfrac{\frac{\sum_j (Y_{t+1} - Y_t)}{(Y_{t+1} - Y_t)}}{A_{jYt+1}}$ <br> $L_{ij}\, \dfrac{\frac{\sum_i (Y_{t+1} - Y_t)}{(Y_{t+1} - Y_t)}}{A_{iYt}}$ | A latent transition is interpreted as existing but apparently inactive and an active transition means that it works or has the capacity to act [54]. |

$D_{ij}$: Number of correctly classified pixels of a particular class; $R_J$: number of reference pixels of the same class; $C_i$: total number of predicted values belonging to a class; N: total number of pixels in the error matrix; $Pr_{(a)}$: total proportion of cells that match in both layers; $Pr_{(e)}$: random hypothetical probability that cells will match in both layers; $P_{+j}$: gain at time two; $P_{jj}$: no coverage change; $P_{j+}$: loss at time two; $G_{ij}$: active transition; $L_{ij}$: latent transition; $Y_{t+1}$: time two; $Y_t$: time one; A: area.

### 2.7. Extraction of Values

All variables were drawn from check points. The extraction of numerical values was carried out using the Point Sampling Tool QGIS software plugin [55]. The model database was generated from this operation.

The extracted numerical values were classified in two phases with a maximum likelihood algorithm [56]. In the first phase, the learning phase, a spectral study of the image was performed for each category at the training points. With this, the spectral response could be related to each category. In the second phase, the probability that each pixel in the images belongs to a certain category was calculated based on the spectral response. The pixel was assigned to the category with the highest probability.

### 2.8. Fitting of Data in the Supervised Learning Model

The database was entered into Salford Predictive Modeler V8.2 software [57], which includes the nonparametric CDT and MARS models. Using a Spearman's rank correlation matrix [58], only the values with a strong correlation (>0.9) were evaluated to reduce the redundant data of the response variable.

The importance of the variables for each of the analyzed algorithms was studied with the purpose of improving their performance. Through post-tuning, the training data were adjusted, maximizing the validation and simplicity of the number of branches in the tree, thus compensating for the lack of backtracking of the induction process [59].

### 2.9. Nonparametric Methods of Classification

#### 2.9.1. CART Decision Tree (CDT)

CDT is an approximation model in which a number of variables established in a nonparametric form are expressed [60]. It establishes an adjustment through recursive binary partitions, in which a successive set of possibilities is established, which give rise to a group belonging to the same characteristics that define its classification. The algorithm is in charge of analyzing the categorical variables that make it possible to form a homogeneous group or create nodes with each other as well as the heterogeneity between each node [61].

The tree is generated from a main node (root) in which all the variables are represented. The algorithm defines the partition of secondary nodes based on increasingly different defined criteria. The separation of these nodes or subsets comprises a classification level. Consequently, if there are new partitions, new secondary nodes are created, but if the data are homogeneous and represent a similar characteristic, then this node becomes a terminal node. The process can be outlined in four phases: tree construction; stopping the tree growth process that constitutes a maximum tree that over-adjusts the information contained in our database; tree pruning, which simplifies the tree by leaving only the most important nodes; and, finally, the selection of the optimal tree with generalization capacity [62,63].

The purity of the nodes must be as high as possible; thus, the CDT method uses the Gini index (1) as a division criterion:

$$g(t) = \sum_{j \neq i}^{i} p\left(\frac{j}{t}\right) p\left(\frac{i}{t}\right) \tag{1}$$

where i and j are the categories of the predictor variable, t is a node, and p is the proportion.

#### 2.9.2. Multivariate Adaptive Regression Splines (MARS)

This algorithm, which is based on recursive partitions and multistage regression, uses spline functions to align data with an arbitrary regression function. It builds this relationship from a set of coefficients and basic functions, which in turn are strongly influenced by the regression of the data [63].

MARS generates cut-off points for the different variables. These points are identified through basal functions, which indicate the beginning and end of a region. The final model

is established as a combination of the generated base functions. To determine these cut-off points, an overestimated model is generated by means of the forward stepwise algorithm. Later, the backward stepwise algorithm is used to eliminate the nodes that contribute the least to the global fit. The algorithm stops when the constructed approximation includes a maximum number of functions set by the researcher [64].

The model can be written as Equation (2):

$$y_t = \int (x_t) = \beta_0 + \sum_{i=1}^{k} \beta_i B(x_{it}) \tag{2}$$

where $y_t$ is the response variable at time t, and $\beta_i$ represents the model parameters for the corresponding variables $x_{it}$, ranging from i = 1, ... , k. The value $\beta_0$ represents the intercept and the base functions $B(x_{it})$ are functions that depend on the corresponding variables $x_{it}$, where each $B(x_{it})$ can be written as $B(x_{it}) = \max(0, x_{it} - c)$ or $B(x_{it}) = \max(0, c - x_{it})$. c is a threshold value and k represents the number of explanatory variables, including interactions of the predictor variables [65].

For the two nonparametric models, 70% of the analyzed objects were used for the learning phase (L) and 30% were used for the validation phase (V).

## 3. Results and Discussion

The variables NDMI, BSI, GSOC, VARI, DEM, and NDMI were those that had the best relationship with the characteristics of the study area. Through their combination in a supervised learning method, it was possible to establish a set of guidelines that allowed the determination of land uses in the Andean páramo.

### 3.1. Spearman's Rank Correlation Matrix—Order Matrix

Table 3 shows the spectral indices that are correlated and provide nonredundant information to the study. Those indices that presented a perfect correlation were eliminated (e.g., SAVI, EVI, BSI, NGRDI, ARVI, GCI, and GNDVI) since they were built on others; specifically, they provided similar data that limited the adjustment of the model and they did not contribute to the decrease in entropy of the resulting information.

**Table 3.** Spearman's rank correlation matrix—order matrix.

|        | NDVI  | VARI  | BSI   | NDMI |
|--------|-------|-------|-------|------|
| NDVI   | 1.00  |       |       |      |
| VARI   | 0.62  | 1.00  |       |      |
| BSI    | −0.75 | −0.84 | 1.00  |      |
| NDMI   | 0.78  | 0.77  | −0.99 | 1.00 |

According to Charles Spearman [58], in his book "General intelligence objectively determined and measured", the correlation between 0.09 and 0.20 is minimal; if the value is between 0.21 and 0.40, it is low; values between 0.41 and 0.60 are moderate; values between 0.61 and 0.8 are good; and values between 0.81 and 1.0 represent a very good correlation. Applying these criteria to our Spearman matrix, the selected spectral indices showed good to very good correlations.

The chlorophyll content (NDVI) reflected the most important direct correlation with the level of water stress of the vegetation (NDMI), while bare soil (BSI) reflected the strongest inversely proportional correlation with the water capacity of the site (NDMI).

The visible vegetation spectrum (VARI) and the chlorophyll content (NDVI) had a good direct correlation with the level of vegetation water stress (NDMI). This is expected since the availability of water is related to the concentration of vegetation.

### 3.2. Analysis of the Variables of Importance

The percentages of importance of the variables (Figure 3) were relatively acceptable considering the difficulties in the study area (e.g., they are dry places, they are difficult to access, and they have complex weather conditions) [1]. Furthermore, it must be considered that the correlation between the variables could have had an impact on the relative evaluation of the importance of the variables but did not affect the performance of the model [43].

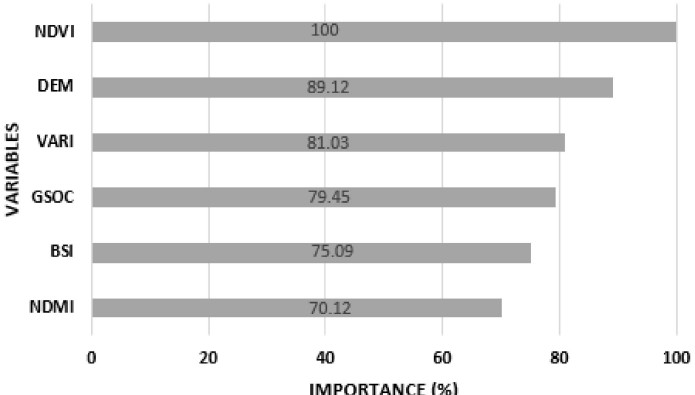

**Figure 3.** Importance of variables.

In the two nonparametric methods, the trends of the variables were the same, ranking in ascending order of importance as follows: NDMI, BSI, GSOC, VARI, DEM, and NDVI.

The NDVI was 100% important within the models, making it an excellent exploratory tool for vegetation classes due to its high sensitivity to different chlorophyll concentrations [66].

The DEM obtained a relative importance of 89.12%, positioning itself as the second relevant variable in the statistical model; it was used to determine that the topographic classes analyzed were considerably related to altitude levels ranging from 3500 m.a.s.l. to 5000 m.a.s.l., and therefore, were related to the temperatures associated with its microclimates that were between 2 °C and 20 °C. In other words, the variable directly influenced the distribution, development, and growth of the studied systems.

The spectral visual atmospheric resistance index (VARI) was relevant, with a value of 81.03% in the analysis. It was not very sensitive to atmospheric effects in the visible range of the spectrum and contributed to adjusting the model from the quality of the plants to the analysis of the growth stages of crops due to their excellent correlation with the nitrogen content [47].

The GSOC variable had a relative importance of 79.45%, which reflects that the distribution of organic carbon in the soil is related to the other parameters that were analyzed.

The bare soil index (BSI) was adjusted to 75.09% importance. Based on this indicator, the model will better discern the soil from areas with little vegetation, quantify its mineral composition, and minimize the influence of humidity, increasing the reliability of the algorithm [67].

The NDMI indicator had a relative importance of 70.12%. It has sensitivity to the absorption of leaf moisture, controlling the water stress of the vegetation cover [51]. Despite occupying the last place in the hierarchical order of the selected variables of importance, this parameter can contribute to monitoring the distribution of water in the ecosystem, leading to the geolocation of areas vulnerable to drought [34].

### 3.3. Precision Assessment of Nonparametric Models

In the learning phase of the confusion matrix (Tables 4 and 5), the CDT algorithm correctly classified 3353 objects (88%) and MARS classified 2955 (81%) out of 3915 training

objects, while in the validation phase, the CDT accurately classified 946 objects and MARS accurately classified 829 out of 1174 control objects.

**Table 4.** Confusion learning matrix (L) 70%, validation (V) 30%. CDT.

| Class | L | V | PF(L) | PF(V) | C(L) | C(V) | Pr(L) | Pr(V) | Gr(L) | Gr(V) | S(L) | S(V) | UA(L) | UA(V) | PA(L) | PA(V) |
|-------|------|------|-------|-------|------|------|-------|-------|-------|-------|------|------|-------|-------|-------|-------|
| PF | 150 | 47 | 117 | 39 | 7 | 3 | 6 | 2 | 19 | 2 | 1 | 1 | 78 | 70.21 | 73.13 | 73.13 |
| C | 700 | 267 | 8 | 13 | 580 | 213 | 75 | 18 | 29 | 15 | 8 | 8 | 82.86 | 79.78 | 83.09 | 83.09 |
| Pr | 1600 | 540 | 16 | 3 | 55 | 33 | 1400 | 436 | 112 | 43 | 17 | 5 | 87.5 | 83.85 | 87.99 | 87.99 |
| Gr | 1290 | 273 | 19 | 4 | 41 | 15 | 101 | 26 | 1110 | 226 | 19 | 2 | 86.05 | 82.78 | 87.06 | 87.06 |
| S | 175 | 47 | 0 | 0 | 15 | 2 | 9 | 2 | 5 | 5 | 146 | 38 | 83.43 | 80.85 | 76.44 | 76.44 |
| Total | 3915 | 1174 | | | | | | | | | | | | | | |

**Table 5.** Confusion learning matrix (L) 70%, validation (V) 30%. MARS.

| Class | L | V | PF(L) | PF(V) | C(L) | C(V) | Pr(L) | Pr(V) | Gr(L) | Gr(V) | S(L) | S(V) | UA(L) | UA(V) | PA(L) | PA(V) |
|-------|------|------|-------|-------|------|------|-------|-------|-------|-------|------|------|-------|-------|-------|-------|
| PF | 150 | 47 | 105 | 30 | 10 | 8 | 11 | 6 | 18 | 3 | 6 | 0 | 70.00 | 63.83 | 65.63 | 65.63 |
| C | 700 | 267 | 14 | 10 | 510 | 185 | 106 | 61 | 68 | 9 | 2 | 2 | 72.86 | 69.29 | 70.83 | 70.83 |
| Pr | 1600 | 540 | 21 | 1 | 110 | 73 | 1200 | 380 | 241 | 82 | 28 | 4 | 75.00 | 70.37 | 79.52 | 79.52 |
| Gr | 1290 | 273 | 20 | 6 | 80 | 22 | 167 | 35 | 1007 | 201 | 16 | 9 | 78.06 | 73.63 | 75.09 | 75.09 |
| S | 175 | 47 | 0 | 0 | 10 | 5 | 25 | 4 | 7 | 5 | 133 | 33 | 76.00 | 70.21 | 71.89 | 71.89 |
| Total | 3915 | 1174 | | | | | | | | | | | | | | |

The analysis of "producer accuracy" (PA) and "user accuracy" (UA; Tables 4 and 5) determined that, in both the learning and validation processes, the nonparametric CDT model performed better than the MARS model in classifying the categories analyzed based on the total number of control points included by the operator for each of the topographic coverages. The same performance trend was reflected in the classification of the categories based on the total number of objects recognized by the program.

The general accuracy, "overall accuracy" (OA, Table 6), indicated that the proportion of objects classified correctly in the CDT model corresponding to the learning phase was 88.00%, with a kappa of 86.51%, and that in the learning phase, it was 83.34% with a kappa of 83.49%. Meanwhile, MARS correctly classified 81.83% with a kappa of 79.86% in the learning process, and 75.46% with a kappa of 74.92% in the validation process.

**Table 6.** Summary of the main precision results of the two models.

| Algorithms | OA (L)% | OA (V)% | KAPPA (L)% | KAPPA (V)% |
|------------|---------|---------|------------|------------|
| **CDT** | 88.00 | 83.84 | 86.51 | 83.49 |
| **MARS** | 81.83 | 75.46 | 79.86 | 74.92 |

The CDT algorithm, through its recursive binary partitioning adjustment mechanism, established an excellent successive set of possibilities for analyzing categorical variables and the ability to form groups or nodes homogeneous among themselves and heterogeneous among nodes from the analyzed characteristics. The multistage regressions established by MARS allowed obtaining reliable functions that aligned the information of the variables towards a reliable supervised learning model. The connection analysis between the variables from the categories evaluated in the two models was good, so it was determined that CDT and MARS are stable and reliable statistical models for this case study.

*3.4. Optimal Model with Higher Accuracy CDT*

From the 3915 objects analyzed, 300 subtrees were randomly created with their corresponding sets of variable characteristics in the study area. The tree with the smallest error had 35 nodes (Figure 4) and included all the variables analyzed in the research based on the categories analyzed in the páramo ecosystem.

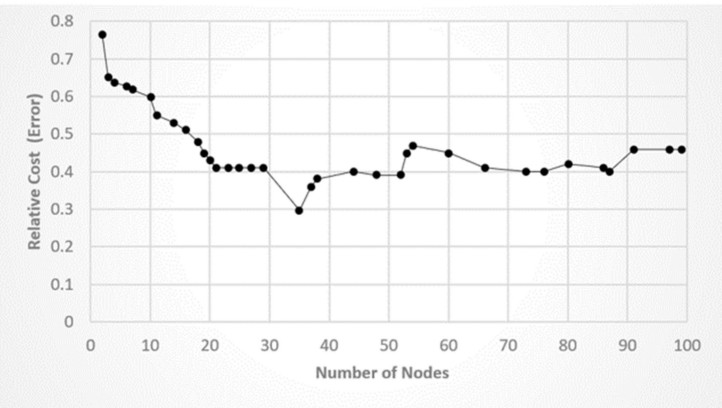

**Figure 4.** Classified error curve.

*3.5. Optimal Model with Higher Accuracy MARS*

The MARS model analyzed 35 basic functions in relation to the variables included in the study (Figure 5). Function 13 was the highest-performing function based on the values fitted by the model and the observer values.

For the analysis of land use changes in páramo ecosystems, the model with the highest degree of accuracy (CDT) was selected.

Figure 6 shows the optimal CDT model; this model provides clear decision guidelines with threshold values.

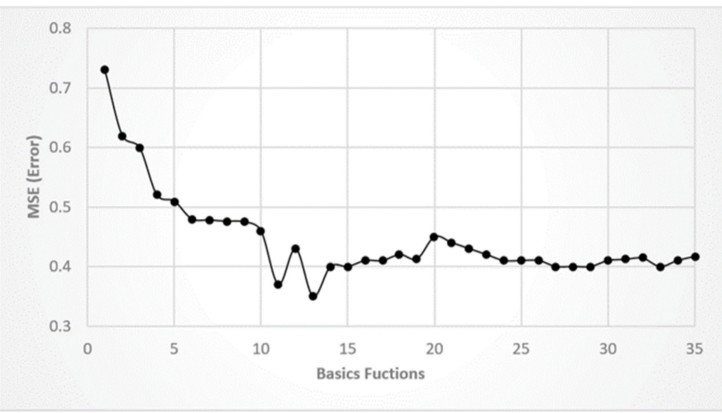

**Figure 5.** Classified error curve.

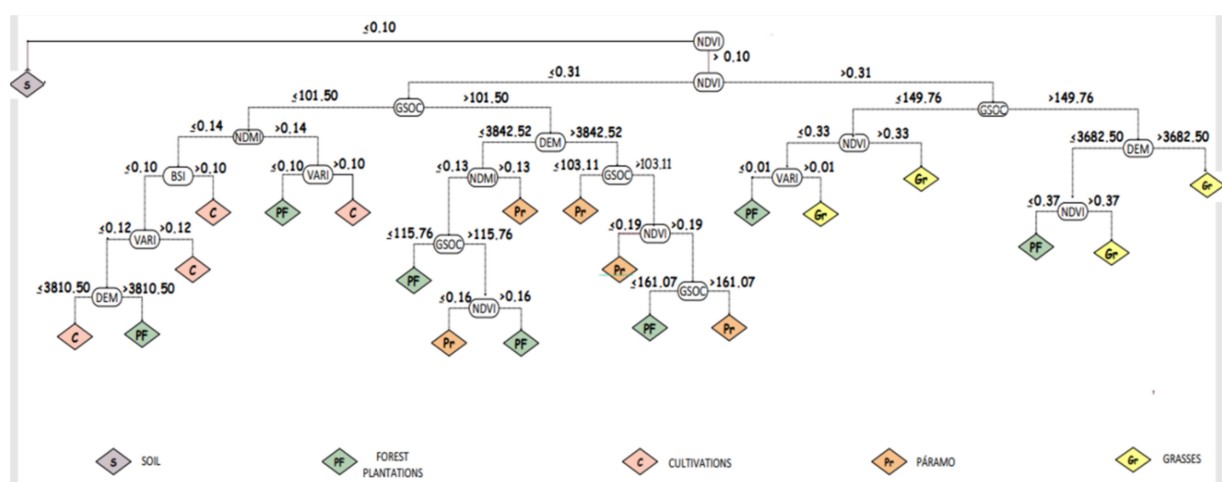

**Figure 6.** The best decision tree of the nonparametric CDT model.

Table 7 shows the conditions to be applied to determine land use coverages in the Andean region studied. Although the model had a high level of accuracy, future researchers could increase its performance considering the thresholds described; in addition, possible variations could cause effects in the different nodes since the whole system is interconnected from the main root, this could be considered a limitation in the method.

**Table 7.** Conditions for the classification of coverage.

| Type | Conditionals | Observation |
|------|--------------|-------------|
| C | NDVI ≤ 0.31, GSOC ≤ 101.50, NDMI > 0.14, VARI > 0.10<br>NDVI ≤ 0.31, GSOC ≤ 101.50, NDMI ≤ 0.14, BSI > 0.10<br>NDVI ≤ 0.31, GSOC ≤ 101.50, NDMI ≤ 0.14, BSI ≤ 0.10, VARI > 0.12<br>NDVI ≤ 0.31, GSOC ≤ 101.50, NDMI ≤ 0.14, BSI ≤ 0.10, VARI ≤ 0.12, DEM ≤ 3810.50 | NDMI is the most important variable in determining crops indicating that crop leaf sensitivity and canopy water stress are directly related to crop development. |
| Pr | NDVI ≤ 0.31, GSOC > 101.50, DEM ≤ 3842.52, NDMI > 0.13<br>NDVI ≤ 0.31, GSOC > 101.50, DEM > 3842.52, GSOC ≤ 103.11<br>NDVI ≤ 0.31, GSOC > 101.50, DEM > 3842.52, GSOC > 103.11, NDVI ≤ 0.19<br>NDVI ≤ 0.31, GSOC > 101.50, DEM ≤ 3842.52, NDMI ≤ 0.13, GSOC > 115.76, NDVI ≤ 0.16<br>NDVI ≤ 0.31, GSOC > 101.50, DEM > 3842.52, GSOC > 103.11, NDVI > 0.19, GSOC > 161.07 | Altitude is one of the variables that significantly determined the distribution of the ecosystem (Pr).<br>The ecosystem can develop above 3842.52 m.a.s.l. |
| Gr | NDVI > 0.31, GSOC ≤ 149.76, NDVI > 0.33<br>NDVI > 0.31, GSOC > 149.76, DEM > 3682.50<br>NDVI > 0.31, GSOC ≤ 149.76, NDVI ≤ 0.33, VARI > 0.01<br>NDVI > 0.31, GSOC > 149.76, DEM ≤ 3682.50, NDVI > 0.37 | The tree determined Gr coverage in two very interesting branches. In one of the branches, the DEM variable is determinant while in the other branch it is not, which leads us to think that the predictive model could be defining one category of natural pasture and another of cultivated pasture. That is, it moves towards natural areas without any control. |
| PF | NDVI > 0.31, GSOC ≤ 149.76, NDVI ≤ 0.33, VARI ≤ 0.01<br>NDVI ≤ 0.31, GSOC ≤ 101.50, NDMI > 0.14, VARI ≤ 0.10<br>NDVI > 0.31, GSOC > 149.76, DEM ≤ 3682.50, NDVI ≤ 0.37<br>NDVI ≤ 0.31, GSOC > 101.50, DEM ≤ 3842.52, NDMI ≤ 0.13, GSOC ≤ 115.76<br>NDVI ≤ 0.31, GSOC > 101.50, DEM ≤ 3842.52 NDMI ≤ 0.13, GSOC > 115.76 > 0.16<br>NDVI ≤ 0.31, GSOC ≤ 101.50, NDMI ≤ 0.14, BSI ≤ 0.10, VARI ≤ 0.12, DEM > 3810.50<br>NDVI ≤ 0.31, GSOC > 101.50, DEM > 3842.52, GSOC > 103.11 > 0.19, GSOC ≤ 161.07 | Forest plantation coverage (FP) has the lowest prediction percentage. It is important to mention that it has the lowest proportioned coverage in the area, so field monitoring could improve its performance. |
| S | NDVI ≤ 0.10 | The NDVI variable was sufficient to determine the ground cover. |

### 3.6. Distribution of the Categories Researched in the Study Area

Figure 7 shows the páramo reported by the map of Ecosystems of Continental Ecuador in 2012 [37] and Figure 8 shows the páramo soil in 2020 and its loss points. Figure 9 shows the alternative land uses by which the páramo was replaced from 2012 to 2020 in relation to the altitudinal levels and levels of organic carbon concentration for each of the comarcas that make up the study area.

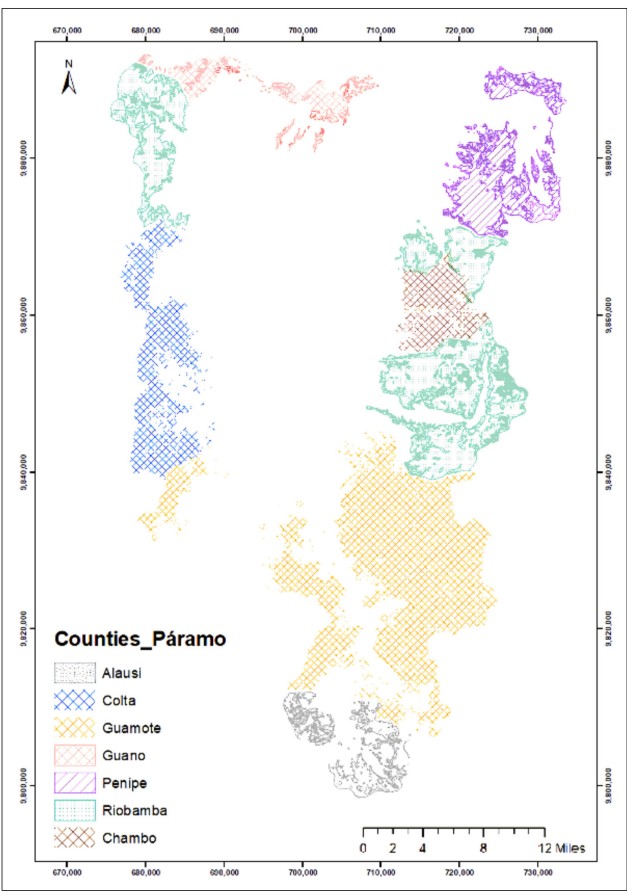

**Figure 7.** Distribution of the páramo according to the counties that comprise the Chambo sub-basin (2012) [37].

The comparison of the 2020 land use map generated from the model that obtained the highest percentage of efficiency (CDT) against the 2012 Map of Continental Ecuador Ecosystems allowed us to evaluate the following aspects: carbon concentrations in relation to the different altitudinal levels and land uses, systematic transitions of cover, gains, losses, exchanges, and net changes of the categories analyzed in the work.

The land uses evaluated in our study are distributed throughout the sub-basin (Figure 9). We found that above 3500 m.a.s.l., the páramo stores 251 to 357 tons C/ha, while forest plantations and grasslands store 81 to 165 tons C/ha. Between 3000 and 3500 m.a.s.l., the páramo ecosystem had a carbon sequestration of 191 to 250 tons C/ha; crops store from 60 to 64 tons C/ha, while grasslands and soils store from 39 to 80 tons C/ha. At altitudes of 2500 to 3000 m.a.s.l., organic carbon concentrations varied in the range of 110 to 190 tons C/ha for páramos, from 20 to 38 tons C/ha for grasslands, and from 10 to 29 tons C/ha for crops.

Table 8 show the changes in the categories in relation to gains, losses, exchanges, and net changes between the land uses of the Map of Ecosystems of Continental Ecuador, MAE (2012) and the Map of Land Uses (2020) generated from the CDT model.

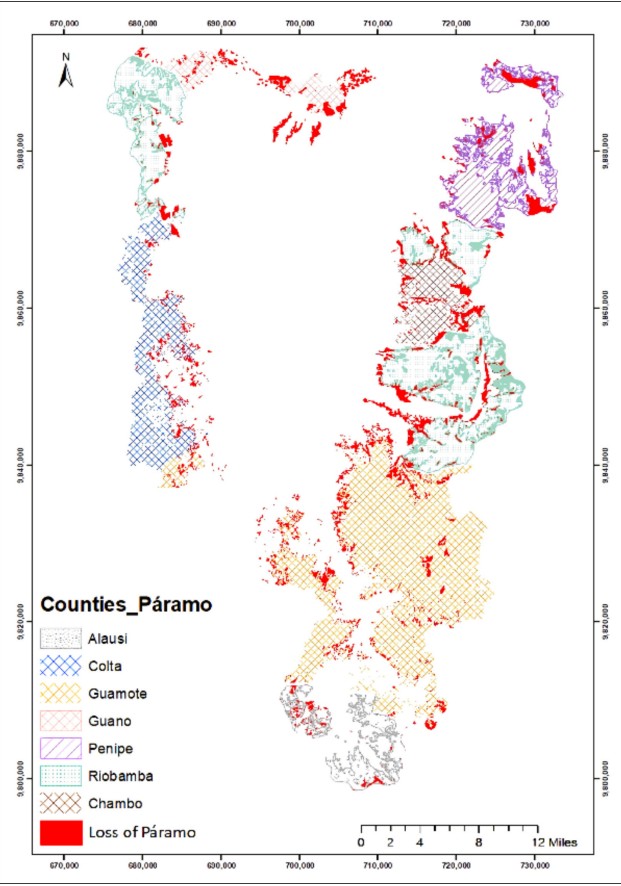

**Figure 8.** Distribution of the páramo according to the counties that comprise the Chambo subbasin (2020).

In 2012, the Ministry of Environment of Ecuador, through the map of ecosystems of continental Ecuador, reported that there were 128,170.48 ha of páramo ecosystem in the Chambo sub-basin [37]. Based on this information and with the data obtained through the land use map elaborated from the CDT model proposed in this research, we found that the overall land use changes in the study area are as follows: the páramo lost 16.65% (21,346.10 ha) of existing lands, while it gained 7.65% from other soil covers. Pastures gained 7.84% while losing 0.82%; crops gained 2.15% and lost 1.52%; forest plantations gained 1.53% and lost 0.98%, and soils gained 5.11% and lost 0.18%.

The interrelation of the cover change indexes indicates that páramo and pasture are the land uses with the greatest transition. Páramo is the land cover with greater loss than gain in relation to the land uses of the other land covers. Overall, the páramo had a loss of 9% in extension during the 8 years from 2012 to 2020.

The loss of the páramo ecosystem is detailed below for each of the counties that make up the Chambo sub-basin. (Table 9).

In ascending order, the loss of páramo in each county is as follows: Alausí 0.61% (785.38 ha), Chambo 1.05% (1343.84 ha), Colta 1.09% (1400.88 ha), Penipe 1.64% (2096.09 ha), Guano 1.73% (2212.49 ha), Guamote 4.18% (5359.40 ha), and Riobamba 6.36% (8148.02 ha).

Riobamba County followed by Guamote are the most affected areas since they are located on the western boundary of the sub-basin [68]. Access to these high-altitude areas is easier, which has allowed the development of agricultural and livestock frontiers to reach the summit line. PF and land cover also reflect a higher concentration in these two counties.

In all the counties, the use of land as pastures represents the greatest impact on the páramo ecosystem, possibly due to policies for price stabilization within the livestock segment, a factor that has provided some stability for producers, making them wager more towards this sector.

The change in the ecosystem from páramo to bare soil may reflect erosion problems because of agricultural practices and inadequate recovery arrangements of natural resources.

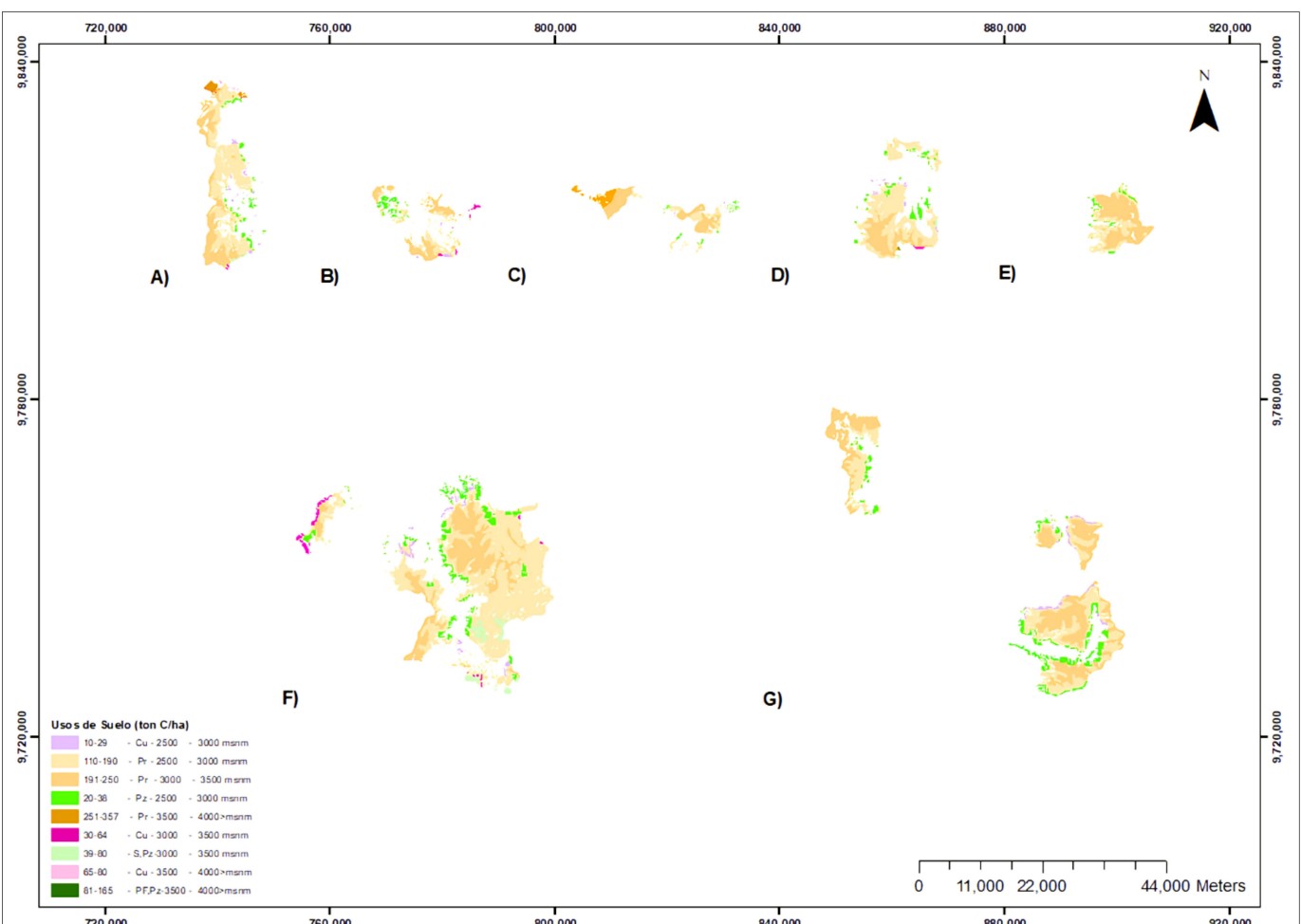

**Figure 9.** Land uses of the páramo ecosystem (2020); ((**A**): Colta; (**B**): Alausí; (**C**): Guano; (**D**): Penipe; (**E**): Chambo; (**F**): Guamote; (**G**): Riobamba; Pr: páramo; Gr: grassland; C: crop; S: soil; and PF: forest plant).

**Table 8.** General index of coverage changes in the study area.

| Coverage | Gains % | Losses% | Exchange% | Net Change% |
|:---:|:---:|:---:|:---:|:---:|
| **Pr** | 7.65 | 16.65 | 12.76 | 7.93 |
| **C** | 2.15 | 1.52 | 3.04 | 0.63 |
| **Gr** | 7.84 | 0.32 | 0.64 | 7.52 |
| **PF** | 1.53 | 0.98 | 1.96 | 0.55 |
| **S** | 5.11 | 0.18 | 0.36 | 4.93 |

**Table 9.** Distribution of land uses in the páramo ecosystem.

| County | UTM Coordinates—Zone 17 Southern Hemisphere | | Pr-MAE (2012) | Loss of PÁRAMO from the Coverage Studied 2012–2020 (Ha) | | | | | Loss of PÁRAMO from the Coverage Studied 2012–2020 (%) | | | | |
|---|---|---|---|---|---|---|---|---|---|---|---|---|---|
| | | | | C | Gr | PF | S | Total | C | Gr | PF | S | Total |
| | X | Y | | | | | | | | | | | |
| ALAUSÍ | 766,363.14 | 9,750,636.59 | 5176.00 | 181.12 | 307.12 | 241.14 | 55.99 | 785.38 | 0.14 | 0.24 | 0.19 | 0.04 | 0.61 |
| CHAMBO | 777,562.97 | 9,805,829.89 | 8220.56 | 172.96 | 418.00 | 107.20 | 645.68 | 1343.84 | 0.13 | 0.33 | 0.08 | 0.50 | 1.05 |
| COLTA | 742,465.20 | 9,799,454.32 | 14,454.90 | 254.01 | 1093.07 | 47.87 | 5.93 | 1400.88 | 0.20 | 0.85 | 0.04 | 0.01 | 1.09 |
| GUAMOTE | 770,637.10 | 9,772,754.27 | 48,481.67 | 593.87 | 2488.48 | 687.05 | 1590.00 | 5359.40 | 0.46 | 1.94 | 0.54 | 1.24 | 4.18 |
| GUANO | 755,657.56 | 9,833,453.54 | 5291.30 | 236.08 | 824.65 | 175.89 | 975.87 | 2212.49 | 0.18 | 0.64 | 0.14 | 0.76 | 1.73 |
| PENIPE | 786,059.27 | 9,823,700.11 | 13,667.04 | 182.86 | 940.97 | 364.46 | 607.79 | 2096.09 | 0.14 | 0.73 | 0.28 | 0.47 | 1.64 |
| RIOBAMBA | 769,782.92 | 9,806,343.65 | 32,879.01 | 1148.02 | 3988.42 | 333.53 | 2678.06 | 8148.02 | 0.90 | 3.11 | 0.26 | 2.09 | 6.36 |
| | | | 128,170.48 | | | | | 21,346.10 | | | | | 16.65 |

### 3.7. Systematic Land Use Change

Tables 10 and 11 detail the systematic transitions of gains and losses of land use changes from the Ecosystem Map of Continental Ecuador (2012) and the Land Use Map (2020) elaborated in the study.

The system transitions in Table 10 indicate that pastures replace páramo, crops, and forest plantations, but do not replace soil. Crops gain and replace forest plantation land uses but do not replace páramo, pasture, and soil. Forest plantations gain but do not replace any land cover.

The transitions in terms of losses represented in Table 11 indicate that páramo and crops lose cover and are replaced by pastures. On the other hand, forest plantations are replaced by crops and pastures.

None of the categories has a zero value in the difference between the values observed by the map and the expected value in the classification, so all changes were interpreted as relevant [69–71].

**Table 10.** Systematic transitions in terms of gains from land use changes in the Chambo sub-basin.

| Coverage | Footprint Size | Strength of the Transition | Interpretation |
|---|---|---|---|
| Pr a C | −0.16 | −0.37 | Cultivation gains, cultivation does not replace páramo. |
| Pr a Gr | 0.11 | 0.02 | Grassland gains, grassland replaces páramo. |
| Pr a PF | −0.81 | −0.95 | Plantation forest gains, plantation forest does not replace páramo. |
| Pr a S | −3.11 | −0.93 | Soil gains, Soil does not replace páramo. |
| C a Pr | −0.03 | −0.33 | Páramo gains, páramo does not replace crop. |
| C a Gr | 0.33 | 0.89 | Grassland gains, grassland replaces crop. |
| C a PF | −0.35 | −7.00 | Plantation forestry gains, forest plantation does not replace cultivation. |
| C a S | 0.11 | 0.58 | Soil gains, soil replaces crop. |
| Gr a Pr | −0.17 | −0.85 | Páramo gains, páramo does not replace grassland. |
| Gr a C | −0.01 | −0.20 | Crop gains, crop does not replace grassland. |
| Gr a PF | −0.01 | −0.09 | Plantation forest gains, forest plantation does not replace grassland. |
| Gr a S | −2.58 | −6.14 | Soil gains, soil does not replaces grassland. |
| PF a C | 0.03 | 4.00 | Crop gains, crop replaces forest plantation. |
| PFa Gr | 0.04 | 0.94 | Grassland gains, grassland replace forest plantation. |
| PF a S | 0.15 | 5.25 | Soil gains, soil replaces forest plantation. |
| S a Pr | −0.23 | −3.29 | Páramo gains, páramo does not replaces soil. |
| S a C | −0.13 | −6.50 | Crop gains, crop does not replaces soil. |
| S a Gr | −0.29 | −0.97 | Grassland gains, grassland does not replace soil. |
| S a PF | −0.46 | −11.50 | Plantation forestry gains, plantation forestry does not replaces soil. |

**Table 11.** Systematic transitions in terms of losses from land use changes in the Chambo sub-basin.

| Coverage | Footprint Size | Strength of the Transition | Interpretation |
|---|---|---|---|
| Pr a C | −0.70 | −0.72 | Páramo loses, crop does not replace páramo. |
| Pr a Gr | 2.87 | 0.75 | Páramo loses, pastizal replaces páramo. |
| Pr a PF | −0.61 | −0.94 | Páramo loses, forest plantation does not replace páramo. |
| Pr a S | −1.56 | −0.86 | Páramo loses, soil does not replace páramo. |
| C a Pr | −1.05 | −0.95 | Crop loses, páramo does not replace crop. |
| C a Gr | 0.49 | 2.33 | Crop loses, grassland replaces crop. |
| C a PF | −0.36 | −9.00 | Crop loses, plantation forestry does not replace crop. |
| C a S | 0.20 | 2.00 | Crop loses, soil replaces crop. |
| Gr a Pr | −2.67 | −0.99 | Grassland loses, páramo does not replace grassland. |
| Gr a C | −0.09 | −0.69 | Grassland loses, crop does not replace grassland. |
| Gr a PF | −0.01 | −0.11 | Grassland loses, plantation forestry does not replace grassland. |
| Gr a S | −2.75 | −11.00 | Grassland loses, soil does not replace grassland. |
| PF a Pr | −0.42 | −0.98 | Forest plantation loses, páramo does not replace forest plantation. |
| PF a C | 0.03 | 1.50 | Forest plantation loses, crop replaces forest plantation. |
| PFa Gr | −0.07 | −0.88 | Forest plantation loses, grassland does not replace forest plantation. |
| PF a S | 0.46 | 11.50 | Forest plantation loses, grassland replaces forest plantation. |
| S a Pr | −0.45 | −0.60 | Soil loses, páramo does not replace soil. |
| S a C | −0.11 | −2.75 | Soil loses, crop does not replaces soil. |
| S a Gr | −0.14 | −0.93 | Soil loses, grassland does not replace soil. |
| S a PF | −0.48 | −15.00 | Soil loses, plantation forestry does not replace soil. |

It was recognized that changes in land use of the natural resource are strongly linked to the economic activities of the inhabitants. This relationship helps to understand that in the studied area, both ecological and social components are complexly related to each other, which is why the páramo ecosystems of the study area should be understood as a socioecological or socioeconomic system [72].

Grasslands are the predominant land uses in the area in terms of profits. Pasture covers replace crops, which could mean that the inhabitants of the sub-basin concentrate their economic activities in the dairy or livestock industry.

The degradation of the páramo ecosystem is significantly greater than its recovery (Table 8), perhaps because native vegetation is completely eliminated from the Andean zones for alternative uses. Soil productivity begins to decrease due to the complex climatological and topographic conditions; after the fourth year, exploitation becomes unsustainable. Productive activities are moved to other Andean areas to take advantage of them for a new period of time [73].

The loss of Andean soils not only interferes with soil yield for productive activities but also causes reduction of carbon storage, decrease in water regulation, loss of native species, deterioration in ecosystem functions, and compromises the ecosystem services of the natural resource for future generations [74]. Therefore, it is important to identify the synergies of land uses.

Carbon concentrations are directly related to alternative land uses. From the land use changes detected in the study and the carbon concentrations determined for each category, it was possible to understand that the loss of páramo causes a loss of carbon sequestration in the Andean ecosystem. The release of organic carbon into the atmosphere contributes to climate change issues [24].

The distribution of organic carbon in the different altitudinal floors was diverse for the analyzed land uses. Carbon sequestration was higher at high altitudinal levels. At higher altitudes, carbon concentrations were higher, perhaps because at higher altitudes the climatological and topographic conditions become even more difficult, which prevents the use of the natural resource by anthropogenic activities.

Based on the evaluation carried out in this work, it is considered necessary to implement timely conservation strategies according to the reality of the geographic area studied. The updating of information on land use changes in the country's páramos is currently deficient or nonexistent. Constant monitoring in these difficult-to-access areas is a real challenge. Natural resource conservation strategies are usually developed based on assumptions or ad hoc information; there is no updated technical information [3].

The product of the fit model contributes to the generation of databases that allow the evaluation of the impact of land use changes in the páramo ecosystem in relation to altitudinal levels and carbon sequestration.

The technical information obtained through the application of the proposed methodology in combination with other social inputs can contribute to the development of more precise programs, strategies, ordinances, or policies for the care of the páramo.

Although the methodology has good results, future researchers may consider the extension of the set of variables included in the study as a limitation to replicate the study. For this research, it was not possible to include variables of another type, for example climatological variables, due to the scarce databases in the study area. For optimal results, we must do a good job during the initial processing of the data (calibration, atmospheric and topographic correction). Finding images with low noise and with moderate atmospheric affectations can be complicated by the complex climatological factors of the area, which could represent a difficulty in applying the method.

The proposed methodology stands out in relation to other studies due to the information extracted for the determination of the thresholds of the variables; the data were obtained from a compilation of 20 Sentinel 2 satellite images whose resolution is excellent to establish a verification of the control points established on the studied coverages; this guaranteed an excellent database for the elaboration of the model and analysis of the importance of variables. All the inputs to replicate the work are low cost, which makes the method an economically accessible tool. Based on the conditionals established in the methodology, the automation of the algorithm will be fast.

It is essential to create sustainable strategies to recover and protect Andean areas. If the protection of the páramo is not taken seriously, the loss of its functionality could generate serious problems in the ecosystem services it provides, for example, in the water supply of its area of influence.

## 4. Conclusions

The variables NDMI, BSI, GSOC, VARI, DEM, and NDVI have important characteristics for the determination of land use in the páramo ecosystem. This set of variables integrated to a statistical method of supervised learning allows monitoring and documenting land use changes of the Andean ecosystem in relation to the rate of degradation,

exchange, net changes, and their relationship with the carbon concentration in its different altitudinal floors.

The CDT model had a performance of 88% and MARS 81.83%, so we conclude that both algorithms reached an efficient accuracy percentage to determine the land uses of the páramo from the variables selected.

We compared the páramo extension in 2012 from the land use report of the Ministry of Environment in Ecuador and the map of land use of the year 2020 generated in our study. We found that 9% less páramo exists in 2020 compared to 2012 (16.65% degradation, 7.65% recovery). We found that pastures are the main replacement of the native vegetation in the degraded páramo.

The interrelation of land cover change indices indicates that land use transitions in the páramo show a greater tendency to degradation than to recovery of the natural resource. Pastures are the main land use replacing native vegetation in the páramo.

The information acquired through the proposed methodology can serve as an input that, in combination with other social and economic inputs, can support the creation of sustainable conservation strategies and policies more in line with the reality of the area.

The methodology is efficient, economical, and easy to deploy, but could be improved by analyzing new spectral indices and adding more detailed GIS vector layers. However, it would be necessary to create them since there are no official databases with this type of information (in the case of Ecuador).

**Author Contributions:** Conceptualization, Y.P. and J.J.d.F.; methodology, Y.P., J.J.d.F., and M.V.; software, Y.P., M.V., and F.C.; validation, Y.P. and J.J.d.F.; formal analysis, Y.P., M.V., J.J.d.F., and F.C.; investigation, Y.P., F.C., and L.Q.; resources, Y.P. and L.Q.; data curation, Y.P. and L.Q. writing—original draft preparation, Y.P.; writing—review and editing, J.J.d.F. and M.V.; supervision, L.Q. and F.C.; project administration, Y.P.; funding acquisition, Y.P. All authors have read and agreed to the published version of the manuscript.

**Funding:** This research received no external funding.

**Institutional Review Board Statement:** Not applicable.

**Informed Consent Statement:** Not applicable.

**Data Availability Statement:** Data sharing is not applicable to this article.

**Acknowledgments:** The authors give their sincere gratitude to The Research and Development Group for the Environment and Climate Change at the Higher Polytechnic School of Chimborazo for its support through the project "Cost benefits analyses from carbon sequestration through conservation to the various options of adaptations and building resilience to climate change through an ecosystem-based adaptation approach in three communities from the High Andean Region of Ecuador".

**Conflicts of Interest:** The authors declare no conflict of interest.

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
