# Peer review of "Identification of a Set of Variables for the Classification of Páramo Soils Using a Nonparametric Model, Remote Sensing, and Organic Carbon"

_sustainability, doi:10.3390/su13169462_

Round 1

Reviewer 1 Report

This is an interesting paper that identifies the land uses in difficult-to-access areas in the Andean paramo in Ecuador. Two nonparametric classification methods have been applied and compared.

The paper is well written. Despite this, there are some points I propose the authors to improve.

Materials and Methods section

The description of the study area is scarce, information like altitude, main land uses, population, protected natural areas would be explained.

The authors should describe the method applied to determine the number of checkpoints; why 3,915 points were selected and the reason of that specific distribution, was a specific sampling method used?

Regarding, the references of the maximum likelihood method I miss that the first authors who developed and applied the method in the 1990s are cited.

The analysis of the importance of the variables should be clarify, it is quite confusing that from the explanation in page six, the results of figure 3 are obtained.

Results and Discussion Section

The authors show interesting results from the application of the nonparametric methods. The final model of the MARS method should be included in the results. The authors claim that the model can be extrapolated to other areas, there are some limitations to do so such as the minimum and maximum value of the data of the variables applied. It would be interesting for the readers to know the limitations of the models.

Finally, the year of the map of figure 6 should be included in its description. And the map in figure 7 is wrong, please provide a map with the right coordinates, (e.g. Alausi appears in the North but it is in the South according to Fig. 6).

And also, please explain what is the land use of the territory that appears in Figure 6 but does not in Figure 7.

Other points to review:

  1. The predictor variable Global Soil Organic Carbon sometimes appears like SOC, others like GSOC. Unify the nomenclature.
  2. Review SAVI formula in table 1, and also, use the same format for all the formulas in this table.
  3. Explain what t means in equation 1
  4. Check equation 2, what do you mean with “yt is the response variable at time t?”
  5. Page 9, the ranking of the variables by importance is not in ascending order according to Figure 3.

Author Response

Dear reviewer,

The authors appreciate your comments and suggestions for improving the article. We detail below how we addressed them in the new revision of the paper.

1.- The description of the study area is scarce, information like altitude, main land uses, population, protected natural areas would be explained.

We have enlarged the description of the study area which now includes: altitude, main land uses, population and main protected natural areas.

2.- The authors should describe the method applied to determine the number of checkpoints; why 3,915 points were selected and the reason of that specific distribution, was a specific sampling method used?

We have included new content explaining that through random sampling using QGIS software, one point for every every 100 ha of land was extracted from the official Map of Ecosystems of Continental Ecuador (2012). This yielded a total of 3580 random control points which we deemed adequate for statistical purposes. We introduced 20% more points for which official data from the Ministerio de Agricultura y Ganadería exists.  As a final step, 381 points were eliminated due to the fact that they were located in areas for which Sentinel-2 images had noisy pixels due to atmospheric effects. After this selection, we were left with a total of 3,915 control points that we used in the study, of which a proportional 70% of the sample was verified in situ.

3.- Regarding, the references of the maximum likelihood method I miss that the first authors who developed and applied the method in the 1990s are cited.

The authors who developed the method in the 1990s have been properly credited

Fisher, R. A. (1921). On the Probable Error of a Coefficient of Correlation Deduced from a Small Sample. Metron,2(3),3–32.

Fisher, R. A. (1991). Statistical Methods and Scientific Inference. Oxford: Oxford University Press.

Fleishman, A. (1978). A method for simulating non-normal distributions. Psychometrika, 43 (4),521-531.

Rubin, D.B. and Schenker, N. (1987). Interval estimation from multiply-imputed data: A case study using census agriculture industry codes. Journal of Official Statistics, 60(3,)375-387.

Dempsted, A.P, Laird, N.M. and Rubin, D.B. (1977). Maximun likelihood from incomplete data via the EM algorithm. Journal of the American Statistical Association, 81(29),41-60

4.- The analysis of the importance of the variables should be clarify, it is quite confusing that from the explanation in page six, the results of figure 3 are obtained.

The analysis of the variables must be understood in conjunction with how the nonparametric methods (CDT, MARS) and the Salford Predictive Modeler V8.2 software work.

To determine the importance of the variables, adjustments were established by means of recursive binary partitions, in which successive sets of possibilities were established, giving rise to a group of variables belonging to the same characteristics, in this case the categorical variable established in the model (land use). From the number of repetitions of the variables to obtain the response variable, the importance of the variables was established.

5.- The authors show interesting results from the application of the nonparametric methods. The final model of the MARS method should be included in the results. The authors claim that the model can be extrapolated to other areas, there are some limitations to do so such as the minimum and maximum value of the data of the variables applied. It would be interesting for the readers to know the limitations of the models.

A figure was included showing the best final function for MARS in relation to the error of all the basic functions established in the model. Since CDT provides better fits, we did not pursue with the MARS model but sticked to CDT.

  1. More spectral and environmental variables could be included to improve the effectiveness of the model for the case study. It was not possible to include a greater number of variables since the official databases of the study area are very scarce. It has been an area little studied and valued for a long time, so getting information of the area was very complex.  Satellite images: It is paramount to do a good job during initial data processing (calibration, atmospheric and topographic correction). Finding images with low noise and moderate atmospheric affectations can be complicated by the climatological factors of the different zones.
  2. Thresholds established in the variables: The model is built from sequentially connected nodes, so if any adjustment is made to the thresholds established for the variables, exponential changes would occur that could lead to a destabilization of the method and decrease its effectiveness.

6.- Finally, the year of the map of figure 6 should be included in its description. And the map in figure 7 is wrong, please provide a map with the right coordinates, (e.g. Alausi appears in the North but it is in the South according to Fig. 6).

The year of figure 6 was placed and a verification and correction of coordinates was performed in figure 7.

7.- And also, please explain what is the land use of the territory that appears in Figure 6 but does not in Figure 7.

Now figure 6 is 7 and figure 7 is 9.

Figure 7 represents the land uses represented in the last official map of the Ecosystems of Continental Ecuador in 2012 and Figure 9 represents the land uses (2020) obtained through the methodology of this study. This is why, at the end of Table 7, a comparative of degradation is established with a focus on the Andean moorlands.

The objective of the methodology is to establish a fast, efficient and economical way to monitor hard-to-reach areas and update information that in turn contributes to decision-making for sustainable development.

Other points to review:

1.- The predictor variable Global Soil Organic Carbon sometimes appears like SOC, others like GSOC. Unify the nomenclature.

The nomenclature was unified in the sections referring to the GSOC variable used in the study. In other sections, such as the introduction, organic carbon is described as part of the literature, so the SOC nomenclature was maintained.

2.- Review SAVI formula in table 1, and also, use the same format for all the formulas in this table.

The SAVI formula was corrected and the format of the formulas was checked.

3.- Explain what t means in equation 1

The description of t has been included.

4.- Check equation 2, what do you mean with “yt is the response variable at time t?”

t represents the response variable in the most optimal approximation constructed by the model.

5.- Page 9, the ranking of the variables by importance is not in ascending order according to Figure 3.

The order of the variables was corrected to maintain the relationship between the information in Figure 3 and the description.

The presentation of the results was improved making the article more understandable; the importance of the information obtained through the method proposed in the study was highlighted; a comparison was made between the data of the official map of Ecosystems of Continental Ecuador of the Ministry of Environment of Ecuador (2012) and the data obtained in the work. Through this comparison, new results were added regarding the main land uses that degrade the páramo ecosystem, speed of degradation and the patterns with which changes occur. This information opens a new way to discuss the conservation status of the Andean landscape and to deepen the work.

The article has been enriched with new references improving the contextualization of the theoretical background and research.

English Changes 

The quality of writing, spelling and language editing has been improved with the support of native English speakers who are highly qualified in editing research literature.

The following corrections have been made:

  1. The article was checked for grammar, syntax, and spelling.
  2. Article usage was corrected according to context.
  3. Changes were made to verb tenses relative to the context of the manuscript.
  4. Modifications were made to ensure proper use of prepositions.
  5. Sentences were restructured to comply with English grammatical rules.
  6. Thoroughly reviewed and corrected the syntax of the text.
  7. Some modifications were made to the text to improve the flow and readability of the article.
  8. Some words were removed and others were included to improve the comprehension of the text.
  9. Technical language was improved according to the field of study.

Reviewer 2 Report

Review of the manuscript Parametrization of Land Uses in the Andean Páramo Through Remote Sensing, Organic Carbon Distribution and Nonparametric Classification Methods

This study aimed to evaluate the performance of a predictive algorithm to monitor and predict the different uses of páramo land using three types of attributes (topographic data, remote data from the Sentinel 2 satellite and, SOC data ) and a categorical variable (land-use coverage). Generally, the article is well written, the structure of the text is correct and the description of the methods is sufficient.

The results are well presented. The Figure 4 is not clear, please improve the quality of this Figure.

Discussion section is missing. In Discussion section authors should refer to the literature mentioned in the introduction section. It is necessary to compare the performance of the proposed methodology with alternatives. It is essential to present how the proposed method performs better than the alternatives.

English language spell check required. The article fits the scope of Sustainability Journal.

Author Response

Dear reviewer,

The authors thank you for your comments, which were very valuable and constructive for the improvement of the work. Below, we detail the improvements made and answer the questions raised.

1.- The Figure 4 is not clear, please improve the quality of this Figure.

Improved quality of figure 4.

2.- Discussion section is missing. In Discussion section authors should refer to the literature mentioned in the introduction section. It is necessary to compare the performance of the proposed methodology with alternatives. It is essential to present how the proposed method performs better than the alternatives.

This study stands out from other alternatives for the following reasons:

  • It presents a method capable of calibrating thresholds and determining variables in a useful algorithm for the study of an area that is difficult to access due to its climatological and topographical conditions, little studied and with scarce availability of databases. The research establishes an important starting point for the creation of algorithms in this type of ecosystem.
  • The thresholds of the spectral variables were established by means of Sentinel 2 images, which have an excellent resolution to identify the coverages studied. Much of the bibliography that was studied prior to the study works with Landsat images, which have a lower resolution and therefore a lower percentage of accuracy.
  • All the variables included in the algorithm have a percentage of more than 70%, which establishes a good performance in the method and reduces the cost of error.
  • The thresholds and variables were determined from the compilation of 20 satellite images, which increases the efficiency of the method, in most cases 1 or 2 images are used.
  • The methodology allows establishing a relationship of land use changes in function of carbon concentrations in the different altitudinal levels with good efficiency.
  • It is possible to document the state of the páramo ecosystem quickly and continuously and retrospective studies can be made.
  • The information that can be obtained through the proposed method allows simple coverage change analyses, but also to know the trends of change.
  • The method presents conditionals for each of the variables that can be automated in a simple way, allowing replication of the study.
  • It does not require a costly investment for replication.
  • It contributes to establish criteria on the vulnerable zones of the ecosystem, which will allow the establishment of more segmented conservation measures.

The advantages of the study have been included in the final discussion and conclusions in more detail.

3.- English Changes

The quality of writing, spelling and language editing has been improved with the support of native English speakers who are highly qualified in editing research literature.

The following corrections have been made:

  1. The article was checked for grammar, syntax, and spelling.
  2. Article usage was corrected according to context.
  3. Changes were made to verb tenses relative to the context of the manuscript.
  4. Modifications were made to ensure proper use of prepositions.
  5. Sentences were restructured to comply with English grammatical rules.
  6. Thoroughly reviewed and corrected the syntax of the text.
  7. Some modifications were made to the text to improve the flow and readability of the article.
  8. Some words were removed and others were included to improve the comprehension of the text.
  9. Technical language was improved according to the field of study.

Reviewer 3 Report

This study deploys familiar methods, well-documented, to test the efficacy of two distinct approaches for identifying ground conditions in the Andean Paramo. Ground-truthing of model "predictions" gives the nod to CDT over MARS.  The title is ironic inasmuch as the intent is portrayed as addressing the parametrization of land uses using two different non-parametric models.  But what is its actual attainment?  Is it useful to inform us that in but one setting CDT achieves a modest improvement over MARS?  How generalizable is such a finding?  Is it to inform us about the Andean Paramo?  If it is there should have been presented a more thorough accounting of its results.  Is it to demonstrate an ability to document land use change?  The authors claim there is evidence of degradation but such would require mappings at two distinct (past) points in time.  If there is in fact degradation then we need to know considerably more.  Lastly, can we conclude on the basis of this work that either model has predictive capabilities?  Well, the authors attempt to calibrate remotely sensed data with a sampling of actual ground conditions, then they apply the model so constructed to "forecast" conditions elsewhere across the paramo.  This is a cross-sectional exercise, and it does have value.  However, any comparison of "present" and "future" conditions would be one of comparative statics.  This most emphatically is different from and less useful than a truly predictive modelling exercise in which we model the forces and factors of change that motivate or energize the evolution of the area.   In the end such efforts as are presented here are "backward looking".  That is, in essence the approach taken here is to take two "pictures", compare the two and infer from their examination the direction of change at some specified level of spatial granularity.  Moreover, though the authors assert the utility of their models in policy formulation, we generally would want to have policy variables, most exogenous, explicitly represented in the model.

As for the subject of carbon sequestration, having read the abstract I had hoped to find a really thorough-going analysis of the subject.  But it appears that this topic was simply one of data convenience.  There is almost nothing here that begins to document or analyze this complex subject.  And yet I will assert that this is perhaps the most important avenue for future exploration and I want to give the authors full encouragement to delve more deeply.

As for the quality of the writing, I wish only to point out numerous problems, none insurmountable.  Overall, the originality of this piece is in its examination of the Andean Paramo, yet it tells almost nothing about this critical landscape.  The methodology is familiar and so is not original.  At the same time the authors have two options.  One is to extract more insight from the work already undertaken by addressing the points I have raised.  The other is to make the Andean Paramo the prime subject.  To do so the team should be broadened to include persons more conversant with high mountain landscape processes, and with policy development particularly regarding carbon sequestration.  

Author Response

Dear reviewer,

The authors appreciate your comments and suggestions for improving the article. We detail below how we addressed them in the new revision of the paper.

  1. The title is ironic inasmuch as the intent is portrayed as addressing the parametrization of land uses using two different non-parametric models.  But what is its actual attainment?  Is it useful to inform us that in but one setting CDT achieves a modest improvement over MARS?  How generalizable is such a finding?  Is it to inform us about the Andean Paramo?  If it is there should have been presented a more thorough accounting of its results.  Is it to demonstrate an ability to document land use change? 

These are all very important observations and have made us rethink how we should present our work.

Indeed, we believe that this paper’s major attainment is the identification of a set of variables which perform well under both non-parametric models, and which allow for the identification of changes in land use specific to the páramo. This has value for Andean moorlands are little studied ecosystems of high natural value but unaccessible due to their climatological and topographic conditions.

Hence, we have modified the title accordingly. It now reads: “Identification of a set of variables for the classification of páramo soils using a non-parametric model, remote sensing and organic carbon”.  Also, less emphasis is put on which predictive model is superior, given than CDT is only marginally better than MARS and in truth both provide good results with the selected variables.

The proposed methodology aims to inform and document the status of the Andean páramo. However, given that we only study one sub-basin in Ecuador, we have removed from the text the statement that the model can be extrapolated to other areas. We have added information regarding the factors that future researchers should take into consideration to extend and apply the model to more global areas. This study establishes an important starting point for further research in the field.

We have also expanded the section of Results so the information is clearer and more exhaustive.

  1. The authors claim there is evidence of degradation but such would require mappings at two distinct (past) points in time.  If there is in fact degradation then we need to know considerably more. 

We can claim evidence of degradation because in the study we compare the paramo at two different epochs: 2012 (based on the Ecosystem Map of Continental Ecuador) against 2020 (based on the map generated from the CDT fitting model).

We have expanded the Results section, where we discuss this degradation, and we now provide more detailed information regarding how each land use has evolved during this timeframe for each county. Also, we have added some paragraphs discussing these results.

  1. Lastly, can we conclude on the basis of this work that either model has predictive capabilities?

The word "predictive", as used in the first draft of the paper was inducive to confusion. Our models are supervised learning algorithms that provide knowledge of the target variable by learning decision rules used from the characteristics of the response variable.

For this reason, we have replaced the word predictive by fitting models or supervised learning models (as in "predictive models" -> "fitting models or supervised learning models").

It is important to mention that in literature these algorithms are known as predictive models since from certain variables they can predict a response variable, however, we agree that when describing the models as "predictive" the expression can be interpreted as future predictions and cause confusion so the above mentioned change was made.

  1. Well, the authors attempt to calibrate remotely sensed data with a sampling of actual ground conditions, then they apply the model so constructed to "forecast" conditions elsewhere across the paramo.  This is a cross-sectional exercise, and it does have value.  However, any comparison of "present" and "future" conditions would be one of comparative statics.  This most emphatically is different from and less useful than a truly predictive modelling exercise in which we model the forces and factors of change that motivate or energize the evolution of the area. In the end such efforts as are presented here are "backward looking". 

That is, in essence the approach taken here is to take two "pictures", compare the two and infer from their examination the direction of change at some specified level of spatial granularity.  Moreover, though the authors assert the utility of their models in policy formulation, we generally would want to have policy variables, most exogenous, explicitly represented in the model.

The method presented in the study is mainly useful to know the current state of the páramo ecosystem and to solve the problem of data scarcity in the area. We don't intend it to have predictive capabilities in the sense that we can’t tell how the paramo will evolve in the future. We are using the model to obtain a set of variables which we can use to assess automatically with high certainty what is the land use in each pixel of Sentinel-2 images. Since the word "predictive" outside of the nonparametric models lingo may induce confusion, we have substituted it (see previous comment).

From the information that can be obtained with the proposed methodology, it is possible to generate important databases to evaluate the state of the páramo ecosystem. The model presented in this work is a good initial support for the evaluation of alternative methods that provide ecosystem information . The updating of information on land use changes in the country's páramos is currently deficient or nonexistent. Constant monitoring in these difficult to access areas is a real challenge. Natural resource conservation strategies are usually developed based on assumptions or ad hoc information; there is no updated technical information

The evaluation, definition and calibration of the thresholds of the variables: topographic, spectral and edaphic distribution of carbon in the area from a nonparametric model that provides efficient results has great value since the final product becomes a useful tool for the study area. The product of the model contributes to the generation of databases that allow us to understand the impact of land use changes in the páramo ecosystem in relation to altitudinal levels. 

Systematic transitions identified in relation to the coverages studied were included in the new revision, and patterns of change were identified as a function of the footprint and strength of the cover changes.

The final objective of the model is not the formulation of policies. The model intends to use the information acquired through the proposed methodology to provide data on the state of the páramo ecosystem, which, in combination with other social and economic inputs, can serve as an input to support the creation of sustainable conservation strategies and policies that are more in line with the reality of the area. The aforementioned was clarified in the text of the article.

  1. As for the subject of carbon sequestration, having read the abstract I had hoped to find a really thorough-going analysis of the subject.  But it appears that this topic was simply one of data convenience.  There is almost nothing here that begins to document or analyze this complex subject.  And yet I will assert that this is perhaps the most important avenue for future exploration and I want to give the authors full encouragement to delve more deeply.

In our work the carbon sequestration variable acts in synergy with the other variables for the purpose of the method.  We aknowledge that a deep analysis on carbon sequestration in the páramo is both a complex and an important subject for further exploration.  We thank the reviewer for this remark since it may indeed prove a fruitful line of future research. For the time being, we have included in the new revision a new paragraph and a map (Fig 9) with the thresholds of carbon concentrations at different altitudinal levels for each land cover. 6.  Andean paramo and critical landscape The final product of the work carried out made it possible to identify the main land uses that degrade the páramo ecosystem, their rate of degradation and the patterns in which changes occur. This information opens a new way to discuss the state of the Andean landscape. Main discussion points included in the study: a)     Relationship of the degradation of the páramo ecosystem with the decrease of environmental services: carbon sequestration, water regulation, soil permeability.b)     Economic activities of the inhabitants and their impact on the loss of the páramo. Losses, gains, exchanges and net changes of the coverages studied.c)     Degradation rate of the páramo and its relationship with soil productivity. Systematic changes of the natural resource. 7.  English Changes 

The quality of writing, spelling and language editing has been improved with the support of native English speakers who are highly qualified in editing research literature.

The following corrections have been made:

  1. The article was checked for grammar, syntax, and spelling.
  2. Article usage was corrected according to context.
  3. Changes were made to verb tenses relative to the context of the manuscript.
  4. Modifications were made to ensure proper use of prepositions.
  5. Sentences were restructured to comply with English grammatical rules.
  6. Thoroughly reviewed and corrected the syntax of the text.
  7. Some modifications were made to the text to improve the flow and readability of the article.
  8. Some words were removed and others were included to improve the comprehension of the text.
  9. Technical language was improved according to the field of study.

The article has been enriched with new references improving the contextualization of the theoretical background and research.

Round 2

Reviewer 1 Report

The manuscript has been improved adequately.

Reviewer 3 Report

The authors are to be congratulated for having undertaken to respond so successfully to my suggestions and those of the other referees.  I am impressed with the results.  Overall it is going to be vitally important for researchers to be able to infer ground conditions at both the regional and global scale as we seek to model earth systems and to recommend remedial actions.  Climate ranks high of course on the growing list of such tasks.  Remote sensing of course can go a long way to address these challenges, but complementary modelling packages when properly ground tested can vastly expand the utility of such approaches.  I recommend acceptance and also wish to encourage the authors to continue their pursuits, one of which should address matters of carbon sequestration.